# Development of core outcome sets and core outcome measures for central visual impairment, visual field loss and ocular motility disorders due to stroke: a Delphi and consensus study

Fiona J Rowe ,[1] Lauren R Hepworth ,[1] Jamie J Kirkham [2]

[1]Institute of Population Health, University of Liverpool, Liverpool, UK
[2]Centre for Biostatistics, Manchester University, Manchester Academic Health Science Centre, Manchester, UK

**Correspondence to**
Prof Fiona J Rowe;
rowef@liverpool.ac.uk

## ABSTRACT

**Objectives** Reporting of research for stroke-related visual impairment is inconsistent. The aim of this study was to define three core outcome sets (COS) and related core outcome measurements (COM) for central visual impairment, visual field loss and ocular motility disorders in stroke research.

**Design** The consensus process consisted of an online three-round Delphi survey followed by a consensus meeting of key stakeholders.

**Setting** UK-wide survey.

**Participants** Stakeholders included orthoptists, occupational therapists, ophthalmologists, stroke survivors and COS users such as researchers, journal editors and guideline developers.

**Outcome measures** For COS development, a list of potentially relevant visual outcomes was created after review of the literature and further grouped into outcome domains. For COM development, a list of potential instruments was created after review of the literature and quality appraised for reliability and validity.

**Results** COS—119 potential outcomes extracted from published literature. Similar assessment outcomes were grouped into 24 outcome domains. Delphi process included 123 participants in round 1, 65 round 2, 51 round 3. Twelve participants attended the consensus meeting with recommended outcome domains for central visual impairment (visual acuity, functional vision, quality of life), visual field loss (visual fields, functional vision, quality of life) and ocular motility disorders (eye alignment, eye movements, functional vision, quality of life). COM—52 test options extracted from the COS outcomes and grouped into 16 domains. Thirteen participants attended the COM consensus meeting. Recommended instruments for measurement of these outcomes include; Logarithm of the Minimal Angle of Resolution visual acuity, cover test, cardinal position eye movement assessments, peripheral visual field perimetry, Visual Function Questionnaire-25.

**Conclusions** COS and COM are defined for vision research for stroke survivors. Their use has potential to reduce heterogeneity in routine clinical practice and improve standardisation and accuracy of vision assessment. Future research is required to evaluate the use of these COS and COM.

## Strengths and limitations of this study

► We report a new core outcome set (COS) and core outcome measures (COM) for stroke-vision research.
► The COS and COM have a break-down for category of visual impairment; specifically, central vision impairment, visual field loss and ocular motility disorders.
► Composite outcome domains were ranked by Delphi and consensus opinion from multidisciplinary experts and stroke survivors.
► This was not an international study; participants were from UK and Ireland.
► We experienced attrition bias across the Delphi rounds.

## INTRODUCTION

Visual impairment is common as a consequence to stroke with a reported incidence of 60% in stroke survivors.[1] Types of visual impairment generally fall into one of four categories including impairments of central vision, eye movements, visual fields and visual perception.[2] The impact of visual impairment on stroke survivors is wide reaching. Visual impairment can increase risk of collisions and falls, impede activities of daily life such as reading and writing, prevent return to work, prestroke activities and hobbies, and driving.[3–6] Such impacts result in loss of independence, increase likelihood of social isolation and results in mood change, anxiety and depression.[5 6] There are several approaches to the management of these visual impairments including visual scanning therapy, prisms, occlusion, spectacles, drugs, surgery, botulinum toxin, exercises or a combination of two or more of the above.[7–9]

A range of systematic reviews in the field of stroke-vision research have identified that there is considerable variation in the

outcomes being measured and reported in primary research studies, which impacts the ability to compare and synthesise outcome results across studies. Moreover, it was noted that there is a paucity of outcome data available on important patient outcomes such as quality of life and functional visual assessment.[7–11]

To mitigate these issues and to increase the relevance of research, a core outcome set (COS) and core outcome measurements (COM) can be developed which represents an agreed standardised set of outcomes and instruments that should be used and reported in all studies for a specific area of health or healthcare.[12 13] A search of the COMET (Core Outcome Measures in Effectiveness Trials) database revealed that there are several studies that have investigated important outcomes for eyes and vision disease; examples include cataracts and glaucoma but none has specifically looked at stroke.[14]

The aim of this study was to achieve consensus on the content of vision research outcomes for stroke survivors. In this study we report the results of a Delphi process and consensus meetings in the development of three COS and COM for stroke-vision research for the categories of impaired central vision, visual field loss and ocular motility disorders.

## METHODS
Development of the COS involved three phases: (1) the generation of a comprehensive list of outcomes; (2) a Delphi survey involving three rounds to gain consensus as to which outcomes are most important and (3) patient and professional consensus meetings to agree on a final COS.

Development of the COM involved three phases: (1) the generation of a comprehensive list of outcome measurement instruments specific to the outcomes in each COS, (2) quality appraisal of the outcome measurement instruments and (3) a consensus process on recommendations for the selection of outcome measurement instruments.

A protocol for the development of this COS project was written by the steering committee, registered in the COMET initiative website (http://www.comet-initiative.org/studies/details/275?result=true) and available as open access (http://pcwww.liv.ac.uk/~rowef/index_files/Page356.html - online supplemental files). When developing this COS, we followed the minimum set of development standards set out by COS-STAndards for Development (COS-STAD) Statement and COnsensus-based Standards for the selection of health Measurement InstrumeNts (COSMIN), and report the results against the COS–STAndards for Reporting (COS-STAR) guideline.[13 15 16]

The full methods for the COS have been fully outlined previously[17] and detail the steering group, patient and public involvement, and stakeholder groups.

### Patient and public involvement
This study was supported by a patient advisory group (the VISable patient and public involvement panel) which provided input to this research study. The VISable panel met on a regular basis during the conduct of the study. Patients contributed to the design of the study and were involved at all stages of the survey and consensus meeting.

### Core outcome set
#### Phase 1: outcome identification
A literature review was conducted to develop a preliminary list of outcomes for the Delphi survey. We undertook an overview of seven systematic reviews of studies/trials reporting vision screening, assessment and treatment of poststroke visual impairment[2 7–11 18–20] and extracted a list of included outcomes. The list was then circulated to the VISable patient and public involvement panel[21] for approval and checking of writing for lay terms as the basis of the online survey development. VISable (a panel of 10 patients/carers) were asked to consider whether any further outcomes should be added, particularly those deemed relevant to patient-reported outcomes. The panel advised that appropriate patient-important outcomes were included already in the list and did not add any further outcomes.

#### Phase 2: Delphi survey
We undertook a prospective consensus study using a Delphi process. SurveyMonkey (SurveyMonkey Inc. Palo Alto, California, USA 2015) was used as the online platform to administer the Delphi process. The survey was piloted for checks of ambiguity and appropriate use of lay language by the steering group and then released live. The full survey was circulated through professional, clinical and research networks in the UK and Ireland to reach stakeholder groups of eye care teams (orthoptists, ophthalmologists), stroke teams (physicians, nurses, therapists) and patients (stroke survivors).

The Delphi survey consisted of three rounds. Rounds remained open for 10 weeks, with regular 2-week reminders sent to those that had partially completed or not completed the survey in order to maximise response rates. All terms had explanatory notes to aid interpretation. All participants were asked to score each outcome in terms of importance and asked to identify any additional outcomes of importance that did not appear in the list of assessments. Additional outcomes added in round 1 were reviewed by the steering group to consider their relevance to the survey and to identify and remove duplicates.

Scoring method: In each round, participants were asked to score the importance of each outcome listed on a nine-point scale (1–3: not important; 4–6: important but not critical; 7–9: critical) as well as an 'unable to score' option. The scale was devised by the Grades of Recommendations, Assessment, Development and Evaluation (GRADE) working group[22] to score the quality of evidence for outcomes in systematic reviews and has been

adopted in other COS development work research using Delphi methods.

Methods of analysis: For each round of Delphi, the results for each outcome were summarised, which included providing the number of participants responding to the survey, as well as the mean, SD and the percentage of participants scoring the outcome at each possible level from 1 to 9.

In round 2, all data from round 1 was analysed and compiled by stakeholder groups; (1) stroke survivors/carers, (2) stroke team professionals and (3) eye team professionals, to allow different perspectives to be considered prior to rerating.[12] The summarised results for each of the three stakeholder groups were provided to all participants in the form of histogram plots. Each participant was also shown their personal original score for each outcome. Participants had the opportunity to rescore based on the summary scores from the three stakeholder groups versus their previous personal scores. All participants were also asked to score on additional outcomes that were identified and added in round 1.

In round 3, the results from round 2 were analysed. Following analysis, the results from each stakeholder group were reviewed by the steering group and judged to be similar in terms of mean responses and percentage spread across the responses of 1–9. Thus, they were presented as one compilation of all responses. Each participant was shown their personal score from round 2 for each outcome along with the summary scores from all other participants, and asked to score each outcome again in terms of importance.

Consensus was defined 'a priori', however, this information was not provided on the Delphi survey. Participants were aware of these cut-off values at the time of attending the consensus meeting and these values were applied overall to responses: 'Consensus in' (ie, consensus that the outcome should be included in the core set) was defined as greater than 70% of participants scoring as 7–9 and less than 15% of participants scoring 1–3. 'Consensus out' (ie, consensus that the outcome should not be included in the core set) was defined as greater than 70% of participants scoring as 1–3 and less than 15% of participants scoring as 7–9. All other combinations were seen as equivocal. The outcomes that were designated as 'consensus in' or seen as 'equivocal' were taken forward and discussed in more detail at the consensus meeting for inclusion into the final COS. Participants were reminded of all outcomes not reaching consensus as part of the Delphi process.

### Phase 3: consensus meeting

All participants from the Delphi survey were invited to contribute to the consensus meeting. Those expressing interest were invited to attend a face to face consensus meeting. All round 3 survey completers were emailed an invitation. Our intention was to ensure all stakeholders were reasonably represented. The format of the meeting included a short study overview which outlined the categories of visual impairment due to stroke and outcomes identified from the outcome identification phase, a presentation containing a summary of the results and number of outcomes reaching consensus from the survey. Each outcome was to be considered in turn, in order of their presentation in the Delphi survey, to ratify these results. Each remaining outcome was then considered in turn with full discussion. Similar to the survey, each outcome was considered as reaching consensus with 70% of participants voting in favour of its inclusion.

Each participant voted on every outcome being asked to vote 'Yes' (this outcome should be included in the COS), 'No' (this outcome should not be included), or 'Unsure' using voting slips. Results were collated after voting for all outcomes was completed and results then presented to the group. Outcomes were retained or dropped when consensus was reached. Discussion and further rounds of voting, restricting the options to 'Yes' or 'No' was undertaken until consensus was reached on all outcomes. All outcomes retained were to be included in the final COS.

## Core outcome measures

### Phase 1: outcome identification

From the same literature review as the COS, a comprehensive list of outcome measurement instruments was generated, with circulation of the list to the VISable panel for review.

### Phase 2: quality appraisal

We undertook a quality appraisal of the available outcome measurements identified from the literature review. For each outcome measurement instrument, we conducted a further review of the medical literature to seek evidence of the use of the instrument in a stroke population. We evaluated the validity of the instrument and feasibility for its use with stroke survivors as per COSMIN guidance.[13] Measurement properties included content validity, internal structure, reliability, measurement error, hypotheses testing, cross-culture validity, criterion validity and responsiveness. Feasibility aspects included comprehensibility, interpretability, ease and type of administration, length of the instrument, completion time, patient mental and physical ability level, ease of standardisation and scoring, type of instrument, cost, equipment required, availability in different settings, copyright and regulatory approval requirements. Quality of evidence was graded from low to high.

### Phase 3: consensus meeting

A subsequent (and separate to COS meeting) consensus meeting was planned for discussion of the COM development. Representatives from the COS Delphi survey were invited to a face-to-face consensus meeting; all round 3 survey completers were emailed an invitation. The intention was to ensure all stakeholders were reasonably represented. The same process as outlined above for COS development was followed for introduction to the process, definition of consensus, voting and compilation of results. The plan was to seek consensus

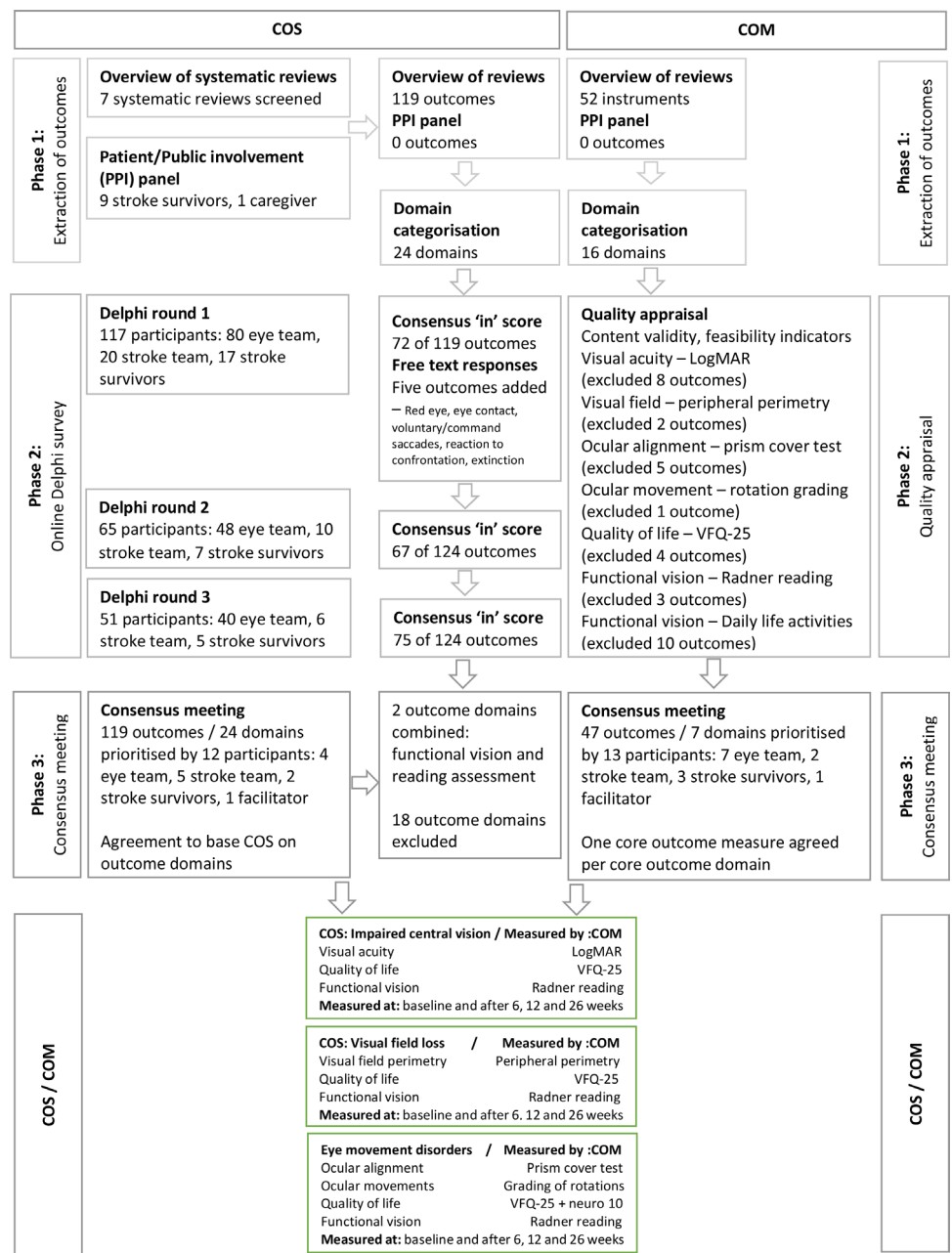

**Figure 1** Flow chart of COS and COM development process. COM, core outcome measurements; COS, core outcome sets; LogMAR, Logarithm of the Minimal Angle of Resolution; VFQ-25, Visual Function Questionnaire 25.

agreement on the inclusion of one instrument for each category of stroke-visual impairment research; specifically, categories of impaired central vision, ocular motility disorders and visual field loss. Consensus agreement was only sought for the agreed COS outcomes. Further, consensus agreement was sought on the timing of measurement.

## RESULTS
Figure 1 outlines a flow chart of the results for number of participants and number of outcomes in the development and agreement of the COS and COM.

## COS process
### Phase 1: outcome identification
We extracted 119 outcomes, many of which were variations on assessments for specific visual functions. For the purposes of clarity during COS development, similar assessments were also combined into outcome domains. This process produced a list of 24 domains (table 1).

### Phase 2: Delphi survey
In total 123 participants registered for round 1 of the Delphi survey. Six registered but did not answer any survey questions. The remaining participants comprised of 79 orthoptists, 20 occupational therapists (stroke team stakeholder group), 17 stroke survivors (stroke survivor/

**Table 1** COS outcome extraction from overview of reviews

| Outcomes for COS | COS domains |
|---|---|
| Case history—open questions | Case history—open questions |
| Case history—specific questions<br>Eye strain, reading difficulty, blurred, altered or reduced vision, visual field loss, awareness of full environment, Oscillopsia, Diplopia, Polyopia, visual hallucinations, altered colour vision, altered movement of objects, depth perception misjudgements, tilted images, distorted images, face/object recognition, clutter difficulty, getting lost, prolongation of images, reverse image size, glare, visual crowding, visual disorientation | Case history—specific questions |
| Case history—carer open questions | Case history—carer open questions |
| Case history—carer specific questions<br>Personal care issues, eyes constantly moving/jerking, missing things to one side, bumping into things, concerns over vision, visual hallucinations, family/friend recognition, difficulty naming objects,<br>getting lost, reading problems | Case history—carer specific questions |
| Case history—previous ocular history | Case history—previous ocular history |
| Case history—glasses wear | Case history—glasses wear |
| Observations—open comments | Observations—open comments |
| Observations—specific features<br>lids, pupils, squint—misaligned eyes, eye movements, turning head to see, closing one eye to see better,<br>misjudging distances, wobbling eyes | Observations—specific features |
| LogMAR charts, Snellen charts, fixation and following observation, vanishing optotype charts, grating charts, near acuity charts, Kay's pictures, Sheridan Gardiner single optotypes, Lea symbols | Visual acuity* |
| Fundus check, retinal photography/OCT | Ocular health* |
| Cover uncover test, alternating cover test,<br>Observations of corneal reflections | Eye alignment position* |
| Nine positions of gaze, Horizontal gaze only,<br>Vertical gaze only, Horizontal and vertical gaze only, Vergence, Saccade movement,<br>Smooth pursuit movement, Optokinetic nystagmus movement, Vestibulo-ocular reflex, Hess/Lees/Harms wall charts | Eye movement assessment* |
| Retinal correspondence, sensory fusion, motor fusion, stereopsis | Binocular vision assessment* |
| Prism cover test, krimsky test, prism reflection test, synoptophore, bruckner test, Maddox rod | Eye alignment measurements* |
| Confrontation, static central perimetry, static peripheral perimetry, kinetic perimetry | Visual field assessment* |
| Line bisection, cancellation task—star, balloon, heart, etc, clock drawing, room/environment description, behaviour inattention test battery | Visual neglect assessment* |
| Observed navigation, reading, eye scanning, walking observations, activities of daily living, self-care, body placement, spatial awareness, mobility observations, writing, hand-eye coordination, visual memory and cognition, visual perception | Functional assessment* |
| Special test, for example, Wilkins, iRest, Radner, Newspaper/magazine, Book | Reading assessment* |
| Vision-related, for example, VFQ25, DLDV; Health-related, for example, SF12; Activity of daily living, for example, IADL; Extended activity of daily living, for example, NEADL | Questionnaires* |
| Visual perception—checklist | Visual perception—checklist* |
| Swinging flashlight test | Pupil assessment* |
| Palpebral apertures, Lid function test | Lid assessment* |
| Pelli-Robson chart, Mars test, VisTech | Contrast sensitivity assessment* |
| Ishihara test, city test | Colour vision assessment* |

COM outcomes indicated in shaded cells.
*Indicates COM domains.
COM, core outcome measurements; DLDV, Daily Living tasks Dependent on Vision; IADL, Instrumental Activities of Daily Living; iReST, International Reading Speed Test; LogMAR, Logarithm of the Minimal Angle of Resolution; NEADL, Nottingham Extended Activities of Daily Living; OCT, Optical Coherence Tomography; SF12, Short Form 12; VFQ25, Visual Function Questionnaire 25.

carer stakeholder group) and 1 ophthalmologist. The orthoptists and ophthalmologist formed the eye team stakeholder group. There were 20 males and 97 females with ages ranging from 18 to 84 years. In round 2, 65 participants completed the survey—an attrition of 44.5% from round 1. These participants comprised 47 orthoptists, 10 occupational therapists, 7 stroke survivors and 1 ophthalmologist. In round 3, 51 participants completed the survey—an attrition of 56.4% from round 1 (21.5% from round 2). These participants comprised of 39 orthoptists, 6 occupational therapists, 5 stroke survivors and 1 ophthalmologist.

Following completion of the survey of 119 outcomes in round 1, five additional outcomes were put forward by the participants for inclusion in round 2: red eye, eye contact, reaction to confrontation, visual extinction and voluntary/command saccades. Consensus 'in' scores were achieved for 72 outcomes. No outcomes scored less than 3. After round 2 of 124 outcomes, no new outcomes were introduced. Consensus 'in' scores were achieved for 67 outcomes. No outcomes scored less than 3. Following round 3 of 124 outcomes, consensus 'in' scores were achieved for 75 outcomes. The Delphi survey reached the following consensus:

► Full consensus for inclusion was obtained for case history asking patients/carers open and specific questions plus checks of previous ocular history and glasses wear. Full consensus for observations, tests of visual acuity, eye alignment, eye movements, binocular vision, alignment measurement, visual fields, visual inattention, reading, lids and pupils, visual perception and functional visual assessment.

► No consensus was reached for eye contact during conversations, use of questionnaires, contrast sensitivity, colour vision, fundus checks and use of retinal photography/optical coherence tomography.

### Phase 3: consensus meeting

The consensus meeting was a 1-day event held in Liverpool, UK with 12 participants comprising 5 occupational therapists (2 with research roles), 3 orthoptists (1 with a research role), 2 patients, 1 Cochrane editor and 1 facilitator. The facilitator did not take part in the voting. The objective of the meeting was to discuss and vote on the Delphi outcomes—the results of the Delphi survey had been provided to participants prior to and during the meeting.

During the consensus meeting, final consensus for research vision assessment was split according to category of visual impairment, (impaired central vision, visual field loss and ocular motility disorders). Participants decided to organise the COS on outcome domains rather than specific outcomes on the basis that many outcomes were similar and were better represented as a grouped outcome domain (see grouping of outcomes in table 1). Two outcome domains were combined because of overlap in outcomes: functional vision and reading assessments.

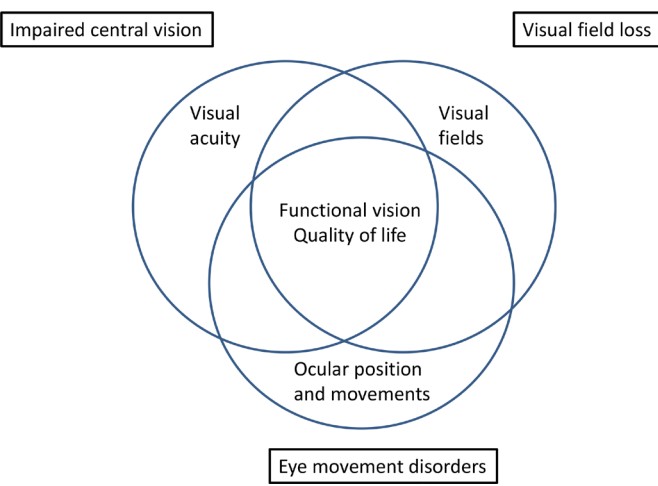

**Figure 2** Core outcome sets consensus.

No consensus could be reached for visual inattention or for visual perceptual disorders. Visual perceptual disorders were considered too heterogenous with so many different types of visual perception issue occurring. It was, therefore, not possible to determine a core set of assessments. Visual inattention was considered too broad and variable a condition with a discussion among consensus meeting participants about context and timing of assessment for stroke survivors with visual attention. Agreement was, therefore, obtained only for categories of impaired central vision, visual field loss and eye movement disorders (figure 2). Impaired central vision consisted of visual acuity, functional vision and quality of life. Visual field loss consisted of visual field perimetry, functional vision and quality of life. Eye movement disorders consisted of ocular alignment, ocular movement, functional vision and quality of life.

### COM process
#### Phase 1: outcome identification
From 119 outcomes identified for the COS, we extracted 52 test options. For the purposes of clarity, similar tests were combined into outcome domains. This process produced a list of 16 domains (table 1).

#### Phase 2: quality appraisal
Quality assessment was conducted for the available outcome measurement instruments considering their measurement properties and feasibility aspects. Table 2 outlines the appraisal summary for selected instruments meeting quality indicators specific to the outcomes selected for the three COS of central vision impairment, visual field loss and ocular motility disorders; Logarithm of the Minimal Angle of Resolution (LogMAR) visual acuity, peripheral perimetry, prism cover test, ocular rotation grading, vision-related quality of life and reading.

#### Phase 3: consensus meeting
The consensus meeting was held as an online event with 13 participants comprising 2 occupational therapists, 7 orthoptists (2 with a research role), 3 patients

**Table 2** Quality indicators of included instruments

| | Visual acuity | Visual field | Ocular alignment | Ocular movement | Quality of life | Functional vision |
|---|---|---|---|---|---|---|
| | LogMAR | Peripheral perimetry | Prism cover test | Rotation grading | VFQ-25 | Radner reading |
| Use in target population | Yes | Yes | Yes | Yes | Yes | Yes |
| Content validity | | | | | | |
| Reliability | + | + | + | + | + | + |
| Responsiveness | + | + | + | + | + | + |
| Internal consistency | + | + | + | + | + | + |
| Structural validity | + | + | + | + | + | + |
| Measurement error | + | + | + | + | + | + |
| Hypothesis testing | + | + | + | + | + | + |
| Criterion validity | + | + | + | + | + | + |
| Cross-cultural validity | + | + | + | + | + | + |
| Quality of evidence | High | High | Moderate | Moderate | Moderate | Moderate |
| Feasibility | | | | | | |
| Patient comprehension | Widely understood | Widely understood | Widely understood | Widely understood | Widely understood | Widely understood |
| Interpretability | Clear usage | Clear usage | Clear usage | Clear usage | Clear usage | Clear usage |
| Ease of administration | High | Relative | High | High | High | High |
| Length of instrument | Small—small number of lines | Moderate—can take time per eye | Small | Small | Moderate | Moderate |
| Completion time | Within minutes | 15–20 min | Within minutes | Within minutes | Relative | Relative |
| Patient mental ability level | Low—use with children | Medium | Low | Low | Medium | Medium |
| Ease of standardisation | High ease | High | High | High | Moderate | Moderate |
| Clinician comprehension | Widely understood | Widely understood | Widely understood | Widely understood | Widely understood | Widely understood |
| Type of instrument | Letter chart | Perimeter | Occluder, target, prisms | Occluder, target | Questionnaire | Reading paragraphs |
| Cost | Low | High | Low | Low | Relative to licence | Low |
| Required equipment | Letter chart | Perimeter | Occluder, target, prisms | Occluder, target | Questionnaire | Book |
| Type of administration | Clinician-led | Clinician led | Clinician led | Clinician led | Clinician-led Self-administered | Clinician led |
| Availability in different settings | Yes | Yes | Yes | Yes | Yes | Yes |
| Copyright | Per company | Per company | Per company | Per company | Per company | Per company |
| Patient physical ability level | Minimal | Relative | Minimal | Minimal | Relative | Relative |
| Regulatory approval | No | No | No | No | No | No |
| Ease of score calculation | High ease | High ease | High ease | High ease | High ease | High ease |

LogMAR, Logarithm of the Minimal Angle of Resolution; VFQ-25, Visual Function Questionnaire 25.

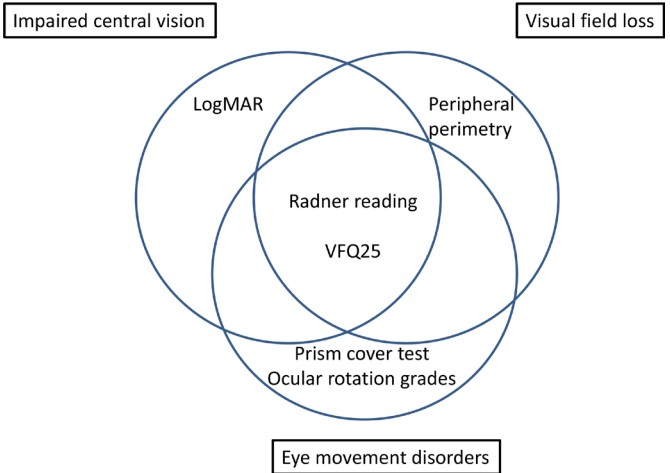

**Figure 3** Core outcome measures consensus. LogMAR, Logarithm of the Minimal Angle of Resolution; VFQ-25, Visual Function Questionnaire 25.

and 1 facilitator. The facilitator did not take part in the voting. The objective of the meeting was to discuss and vote on the COM—the results of a comprehensive list of outcome measurements and their quality appraisal had been provided to participants prior to and during the meeting. Each participant was invited to make queries for each instrument and seek clarification of any part of the quality appraisal. Agreement was requested from each participant on inclusion of each selected instrument.

During the consensus meeting, final consensus for research vision assessment was split according to type of measurement (figure 3). Overall excluded instruments and reasons for exclusion are outlined in table 3. Recommendation for inclusion was obtained for;

► LogMAR test (impaired central vision). This test may require the use of a matching card because of communication or cognitive issues.

► Peripheral perimetry (visual field loss). Specifically, peripheral visual field testing was recommended over central visual field testing to acknowledge the need to detect presence of peripheral visual field loss outside the central 30° and detection of change occurring in the peripheral visual field over time. Kinetic perimetry was recommended with the addition of static points to check the integrity of the visual field within the peripheral boundary. Where static-only programmes were available, a recommendation was made to use the Esterman programme which is most commonly available regardless of equipment type. A recommendation was made to use the binocular Esterman programme as the minimum core assessment but with the addition of monocular Esterman fields for each eye where possible dependent on patient ability.

| Table 3 | Reasons for exclusion of instruments |
| --- | --- |
| **Instrument/measurement** | **Exclusions** |
| Case history | Non-standardised and subjective |
| Observations | Non-standardised and subjective |
| Visual acuity (non-LogMAR options), for example, Snellen charts; Fixation and following observation; Vanishing optotype charts; Grating charts; Near acuity charts; Kay's pictures; Sheridan Gardiner single optotypes; Lea symbols | Inconsistencies in measurement properties |
| Ocular position, for example, Krimsky test; Prism reflection test; Synoptophore; Bruckner test; Maddox rod | Inconsistencies in measurement properties and inclusion of subjective examiner judgments |
| Ocular movements, for example, Hess/Lees/Harms wall charts | Insufficient widespread access and insufficient assessment of peripheral eye movements |
| Visual fields: Confrontation; Central perimetry | Subjective examiner assessment and/or insufficient assessment of the peripheral field of vision beyond 30 degrees |
| Reading, for example, IReST; Wilkins; Vocational reading charts | Reading sections set at specific font size Non-adaptable to low vision patients |
| Activities of daily living (ADL), for example, ADL; Assessment of Motor and Process Skills; Frenchay Activities Index; Functional Independence Measure; Multiple Errands Test; Reintegration to Normal Living Index; Stroke Impact Scale; Assessment of Life Habits; Activity Card Sort; Modified Rankin Scale | No one specific test targeting vision-related activities of daily living |
| Vision-related quality of life, for example, Veterans Low Vision Visual Function Questionnaire; Activity Inventory; Daily Living Tasks Dependent on Vision; Self-Reported Assessment of Functional Visual Performance | Inconsistencies in measurement properties |

LogMAR, Logarithm of the Minimal Angle of Resolution.

► Prism cover test and ocular rotation grading (ocular movement disorders). Ocular alignment determined by the cover test and measured by prism cover test. Ocular rotation gradings specifically in nine positions of gaze and assessed using saccadic and smooth pursuit movements alongside vergence (convergence and divergence). Recommendation to incorporate vestibulo-ocular reflex movements and/or optokinetic nystagmus where relevant (eg, if smooth pursuit movements cannot be fully examined).

► Visual Function Questionnaire 25 (VFQ-25) questionnaire (vision-related quality of life).[23] Recommendation to add neuro-10 supplement when used for ocular motility COS but not appropriate for use in impaired central vision or visual field loss COS.

► Radner test (functional vision: reading impairment).[24] Recommendation to capture additional impact to reading by non-visual issues such as presence of aphasia.

No consensus could be reached for functional vision assessment incorporating activities of daily living. Functional visual assessment was considered too heterogeneous for types of assessment in relation to visual function versus general function, and therefore, it was not possible to determine one, or even a small number, of measurement tools. Consensus was reached that the aspect of functional vision with respect to activities of daily living warrants further research and should consider the issues identified during the Delphi process and focus groups, including; observed navigation, eye scanning, walking observations, activities of daily living, self-care, body placement, spatial awareness, mobility observations, writing, hand–eye coordination, visual memory and cognition and visual perception.

The final stage of the consensus meeting was to consider the time points at which the COM should be made as a minimum. Consensus was achieved for a recommended measurement timescale of baseline and follow-up periods of 4 weeks, 12 weeks and 26 weeks, for all three COS. There was a recommendation that consideration should be given to recruitment in acute versus chronic stages of stroke given that rapid changes may occur in visual function within the first month of stroke onset.

## DISCUSSION

A COS is an agreed minimum set of outcome measures that should be reported. By reporting a minimum set of measures, this reduces the heterogeneity of outcomes across studies, which in turn supports future evidence synthesis.[12 15 16] Once a COS is defined, it is important to achieve consensus on how the outcomes should be measured.[13] COM define how to measure core outcomes and are the instruments selected to measure each core outcome.[13]

We have previously used Delphi and consensus process methods to develop test batteries for general clinical vision screening and assessment for poststroke visual impairment.[17] The purpose of this study was to agree on a COS and COM specific to use in stroke-vision research, distinct from routine clinical screening and assessment practice. After round 3 and the consensus meeting, stakeholders agreed with the proposed COS for vision research for three categories of visual impairment, consisting of three outcome domains for impaired central vision, three for visual field loss and four for eye movement disorders. Core among all categories were the domains of functional vision and quality of life. It is important to acknowledge these COS do not exclude further visual assessments which should be added as appropriate and relevant for the individual research study. Indeed, it is essential to ensure that the multiple visual morbidities that arise due to stroke, are also properly recognised, sought and reported, despite not being selected as core elements.

Generally, one instrument is selected for each COM.[13] For assessment of impaired central vision, the selected instrument to test visual acuity was a logMAR chart, for visual field loss, peripheral perimetry was selected and for eye movement disorders, the selected instrument was the prism cover test when measuring ocular alignment, and grading of ocular rotations for measuring eye movement. For assessment of vision-related quality of life, the VFQ-25 questionnaire was selected and for functional vision assessment specific to reading, the Radner reading test was selected. It was not possible to reach consensus for measurement of other aspects of functional vision. There was unanimous agreement that further research is required to evaluate measurements of activities of daily living in relation to vision vs general function with specific aspects to be considered such as observed navigation and mobility, spatial awareness, self-care and writing. Quality appraisal for COM considered documentation of the use of the instrument in the target population, content validity, quality of evidence, and feasibility of use of the instrument.[24–49] Timing of these measurements were recommended at minimum time points of baseline and follow-up at 4, 12 and 26 weeks.

There are a number of strengths for this study. A COS has been produced for stroke-vision research with a break-down for category of visual impairment; specifically, central vision impairment, visual field loss and ocular motility disorders. This addresses a gap in evidence-based practice for poststroke visual impairment. The COS is composed of outcome domains which were ranked by Delphi and consensus opinion from multi-disciplinary experts and stroke survivors. Furthermore, they are based on existent outcomes for which there are assessment instruments (ie, COM) that are easily accessible for implementation, and accepted as validated clinical measures. Although these COS and COM were developed for stroke, there is potential for their use with other types of acquired brain injury causing visual impairment.

There are limitations to this study. Despite circulation of the survey across a wide range of networks, not all members of eye care and stroke teams took part despite invitation, for example, lack of optometry, neuropsychology input,

ophthalmology/neuro-ophthalmology. Participants were from UK and Ireland and largely represented views and practices within the National Health Service. Thus, assessments are those used clinically in these countries. However, we are aware that the assessment outcome domains selected for the COS and instruments selected for the COM are those that are widely used internationally in clinical and research settings as evidenced from our initial literature review.[2 8 10] Indeed, the quality appraisal of outcome measurement instruments involved many studies undertaken in countries other than the UK.[24–49] We experienced attrition bias across the Delphi rounds. The attrition rate at round 2 was 44.5% and, by round 3, was 56.4% from the initial sample. High attrition rates, however, are common using Delphi methods and our rates can be considered within an acceptable range based on those reported for other COS developments.[50] Nevertheless a larger sample would have been preferable. Researcher bias is also a potential limitation. We aimed to limit this by providing a summary of results across all rounds of the Delphi survey and with final decisions left to the consensus meeting with experts. There can be risks from using Delphi methods in which participants can have very disparate views of each outcome. However, we sought a wide variety of participants across a number of stakeholder groups to achieve greater consistency in responses and balance potential outlier responses. Further, this core outcome development process included a final stage of consensus meetings such that decisions were not purely made from the Delphi responses. Although we initially included outcomes for visual neglect/inattention and visual perception, it was not possible to reach consensus for a COS for these outcomes through the Delphi survey or consensus process. A detailed discussion took place during the consensus process that highlighted the specific issues for these outcomes and, in particular, the issue of context of assessment (where, when) and type (personal/peripersonal/extrapersonal) of visual neglect, and significance of heterogeneity for forms of other visual perceptual disorders. Further it was not possible to reach consensus for a COM for functional vision specific to activities of daily living. Discussion during the consensus meeting highlighted that no specific instrument/tool for activities of daily living focused on functional vision and that this represented a gap in evidence requiring further research. While visual inattention, visual perception disorders and functional visual assessment were not included in the COS and COM developed in this study, these are topics that warrant further research to target core outcomes and measurements. These disorders can be debilitating consequences of stroke affecting the visual regions of the brain and we advocate for further research on development of COS and COM as being essential for these disorders.

## CONCLUSIONS

This study reports the use of Delphi and consensus methods in the development of a COS and COM for stroke-vision research. Impaired central vision and visual field loss each comprised three outcome domains and eye movement disorders comprised four outcome domains. Each COS outcome domain has one recommended outcome measurement instrument except for functional vision relating to activities of daily living for which no consensus could be reached. These COS and COM will facilitate standardisation of stroke-vision assessment in order to reduce heterogeneity in assessment in future research. Further research is now required to evaluate the use of these COS and outcome measures.

**Acknowledgements** We thank all participants involved in the Delphi survey and consensus meetings. We also thank our clinical and patient colleagues who contributed to the study steering group.

**Contributors** FJR is the guarantor for this study. She provided oversight for the study and led the writing of the paper. FJR, LH and JJK contributed to data collection, reviewing the draft paper and approving the final version.

**Funding** This article/paper/report presents independent research funded by the National Institute for Health Research (NIHR: CDF-2012-05-126).

**Disclaimer** The views expressed are those of the authors and not necessarily those of the NHS, the NIHR or the Department of Health.

**Competing interests** None declared.

**Patient and public involvement** Patients and/or the public were involved in the design, or conduct, or reporting, or dissemination plans of this research. Refer to the Methods section for further details.

**Patient consent for publication** Not applicable.

**Ethics approval** This study had institutional ethical approval (Institute of Population Health Ethics Committee; Ref-1415-040). Participants gave informed consent to participate in the study before taking part.

**Provenance and peer review** Not commissioned; externally peer reviewed.

**Data availability statement** Data are available on reasonable request. The Delphi dataset is available from the lead author on reasonable request.

**ORCID iDs**
Fiona J Rowe http://orcid.org/0000-0001-9210-9131
Lauren R Hepworth http://orcid.org/0000-0001-8542-9815
Jamie J Kirkham http://orcid.org/0000-0003-2579-9325

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
