## [Reviewer comments · BMJ Open]

ARTICLE DETAILS

TITLE (PROVISIONAL)	Development of core outcome sets and core outcome measures for central visual impairment, visual field loss and ocular motility disorders due to stroke: a Delphi and consensus study.
AUTHORS	Rowe, Fiona; Hepworth, Lauren; Kirkham, Jamie J.

VERSION 1 – REVIEW

REVIEWER	Gordon Dutton Glasgow Caledonian University, Vision Sciences
REVIEW RETURNED	06-Nov-2021

GENERAL COMMENTS	This study reports the development of a core outcome set and consensus methods of describing the ophthalmic consequences of stroke in future research. The authors comprehensively tabulate potential visual consequences of stroke and hone this down to a set of measurable consequences and measures. The heterogeneity of outcomes and their interplay is well presented. The risk of honing this list down to core outcomes and measures is that this comprehensive range of manifestations could become lost sight of when the outcome of this Delphi study is utilised for research. If the authors could highlight that it is essential to ensure that these multiple visual morbidities are also properly recognised, sought and reported, despite not being selected as core elements, then it is this referee's contention that the paper would be enhanced, and this could avert future potential inter-professional criticism. The final paragraph of page 11 indicates that 'visual perceptual disorders were considered too heterogeneous' and 'visual inattention was considered too broad and variable', precluding the determination of core sets of assessments for these categories of stroke related disordered vision, so agreement was not gained for their inclusion in the development of core outcome sets or measures, in the context of this study. The same can be said for functional visual assessment (P14 line 59). In the second paragraph of the discussion the authors highlight that the absence of inclusion of these categories does not preclude such data from being included in future research. Disorders of visual perception and associated lack of visual attention and impaired functional vision can be debilitating consequences of stroke affecting the visual regions of the brain. For this reason the additional element at the end of the discussion indicating the need for further work (along the lines of approaches taken by this paper) to identify how to ensure that these fundamental elements are not ignored by virtue of not being included the COS and COM recommendations being given, is an essential element of this paper.
--

	P12 Line 44: The number of staff and range of professions attending the consensus meeting could arguably be considered to be somewhat limited. For example no ophthalmologists nor optometrists were described as being present, which has the potential to limit the utility of the Delphi exercise carried out. Extension of the list on the first line of page 17 to included ophthalmologists or neuro-ophthalmologists (at least for the consensus meeting) is warranted. P13 Lines 44 and 49: The VFQ-25 tests and the Radner test are not necessarily well known by the target readership of vision professionals. They therefore merit appropriate referencing. Minor edits: P4 Line 48: ...none have specifically looked at stroke – should read – ...none has specifically looked at stroke (none, meaning not one, is singular) P8 Line 38: Each outcome was be considered in turn – should probably read – Each outcome was to be considered in turn P10 Line 40: comprised of 47 orthoptists – should read – comprised 47 orthoptists P10 Line 53: extinction – should probably read – visual extinction P13 Line 25: ...to use binocular Esterman programme – should read - ...to use the binocular Esterman programme
--	---

REVIEWER	Kimberly Hreha University of Texas Medical Branch at Galveston
REVIEW RETURNED	22-Nov-2021
GENERAL COMMENTS	This is a very well written manuscript that describes an important research project. Looking forward to reading the future study where you evaluate the use of these COS and COMs.

VERSION 1 – AUTHOR RESPONSE

Reviewer: 1

The authors comprehensively tabulate potential visual consequences of stroke and hone this down to a set of measurable consequences and measures. The heterogeneity of outcomes and their interplay is well presented. The risk of honing this list down to core outcomes and measures is that this comprehensive range of manifestations could become lost sight of when the outcome of this Delphi study is utilised for research.

If the authors could highlight that it is essential to ensure that these multiple visual morbidities are also properly recognised, sought and reported, despite not being selected as core elements, then it is this referee's contention that the paper would be enhanced, and this could avert future potential inter-professional criticism.

We have highlighted this issue in the Discussion as suggested – thank you.

The final paragraph of page 11 indicates that 'visual perceptual disorders were considered too heterogeneous' and 'visual inattention was considered too broad and variable', precluding the determination of core sets of assessments for these categories of stroke related disordered vision, so agreement was not gained for their inclusion in the development of core outcome sets or measures, in the context of this study.

The same can be said for functional visual assessment (P14 line 59). In the second paragraph of the discussion the authors highlight that the absence of inclusion of these categories does not preclude

such data from being included in future research. Disorders of visual perception and associated lack of visual attention and impaired functional vision can be debilitating consequences of stroke affecting the visual regions of the brain. For this reason the additional element at the end of the discussion indicating the need for further work (along the lines of approaches taken by this paper) to identify how to ensure that these fundamental elements are not ignored by virtue of not being included the COS and COM recommendations being given, is an essential element of this paper.

Thank you – we have further highlighted this recommendation at the end of the Discussion.

P12 Line 44:

The number of staff and range of professions attending the consensus meeting could arguably be considered to be somewhat limited. For example no ophthalmologists nor optometrists were described as being present, which has the potential to limit the utility of the Delphi exercise carried out. Extension of the list on the first line of page 17 to included ophthalmologists or neuro-ophthalmologists (at least for the consensus meeting) is warranted.

We agree and have added ophthalmology / neuro-ophthalmology. These professional groups were invited to participate but none came forward.

P13 Lines 44 and 49:

The VFQ-25 tests and the Radner test are not necessarily well known by the target readership of vision professionals. They therefore merit appropriate referencing.

References added.

Minor edits:

P4 Line 48: ...none have specifically looked at stroke – should read – ...none has specifically looked at stroke (none, meaning not one, is singular)

P8 Line 38: Each outcome was be considered in turn – should probably read – Each outcome was to be considered in turn

P10 Line 40: comprised of 47 orthoptists – should read – comprised 47 orthoptists

P10 Line 53: extinction – should probably read – visual extinction

P13 Line 25: ...to use binocular Esterman programme – should read - ...to use the binocular Esterman programme

Thank you – these edits have been corrected.

Reviewer: 2

This is a very well written manuscript that describes an important research project. Looking forward to reading the future study where you evaluate the use of these COS and COMs.

Thank you.

We thank the editor and reviewers for these positive comments and suggestions.

VERSION 2 – REVIEW

REVIEWER	Gordon Dutton Glasgow Caledonian University, Vision Sciences
REVIEW RETURNED	23-Dec-2021

GENERAL COMMENTS	This re-submission has satisfactorily addressed each of the issues highlighted by this reviewer. Suggested minor edits: Page 3 final line: groped – should read – grouped Page 19 final line of penultimate paragraph: we advocate for further research on development of COS and COM as essential for these disorders. – could better read- we advocate for further research on development of COS and COM as being essential for these
---

	disorders.
--	------------

VERSION 2 – AUTHOR RESPONSE

Reviewer: 1

Page 3 final line: groped – should read – grouped

This typo has been corrected.

Page 19 final line of penultimate paragraph: we advocate for further research on development of COS and COM as essential for these disorders. – could better read- we advocate for further research on development of COS and COM as being essential for these disorders.

Of course – amended.